# The Role of ATF3 in Neuronal Differentiation and Development of Neuronal Networks in Opossum Postnatal Cortical Cultures

**DOI:** 10.3390/ijms23094964

**Published:** 2022-04-29

**Authors:** Antonela Petrović, Jelena Ban, Matea Ivaničić, Ivana Tomljanović, Miranda Mladinic

**Affiliations:** Laboratory for Molecular Neurobiology, Department of Biotechnology, University of Rijeka, 51000 Rijeka, Croatia; antonela.petrovic@biotech.uniri.hr (A.P.); jelena.ban@biotech.uniri.hr (J.B.); ivanicic.matea@gmail.com (M.I.); ivana.tomljanovic1@uniri.hr (I.T.)

**Keywords:** opossums, neuronal differentiation, neuroregeneration, neurospheres

## Abstract

Activating transcription factor 3 (ATF3), a member of the ATF/cAMP response element-binding (CREB) family, is upregulated by various intracellular and extracellular signals such as injury and signals related to cell proliferation. ATF3 also belongs to the regeneration-associated genes (RAG) group of transcription factors. RAG and ATF/CREB transcription factors that play an important role in embryonic neuronal development and PNS regeneration may also be involved in postnatal neuronal differentiation and development, as well as in the regeneration of the injured CNS. Here we investigated the effect of ATF3 in differentiation, neural outgrowth, network formation, and regeneration after injury using postnatal dissociated cortical neurons derived from neonatal opossums (*Monodelphis domestica*). Our results show that RAG and ATF genes are differentially expressed in early differentiated neurons versus undifferentiated neurospheres and that many members of those families, ATF3 in particular, are upregulated in cortical cultures obtained from younger animals that have the ability to fully functionally regenerate spinal cord after injury. In addition, we observed different intracellular localization of ATF3 that shifts from nuclear (in neuronal progenitors) to cytoplasmic (in more mature neurons) during neuronal differentiation. The ATF3 inhibition, pharmacological or by specific antibody, reduced the neurite outgrowth and differentiation and caused increased cell death in early differentiating cortical neuronal cultures, suggesting the importance of ATF3 in the CNS development of neonatal opossums. Finally, we investigated the regeneration capacity of primary cortical cultures after mechanical injury using the scratch assay. Remarkably, neonatal opossum-derived cultures retain their capacity to regenerate for up to 1 month in vitro. Inhibition of ATF3 correlates with reduced neurite outgrowth and regeneration after injury. These results indicate that ATF3, and possibly other members of RAG and ATF/CREB family of transcription factors, have an important role both during cortical postnatal development and in response after injury.

## 1. Introduction

The inability of the adult mammalian central nervous system (CNS) to regenerate after an injury is one of the major problems in the treatment of CNS injuries and neurodegenerative diseases. One possible strategy for successful CNS regeneration is to recapitulate the events that normally occur during neuronal development since many molecules involved in neuronal plasticity are thought to be crucial both for development and regeneration [1,2]. For example, regeneration-associated genes (RAG) and activating transcription factors (ATF)/cAMP response element-binding (CREB) transcription factors that play an important role in embryonic neuronal development and peripheral nervous system (PNS) regeneration may be involved in postnatal development and regeneration of injured adult CNS [2,3,4,5].

A common feature of ATF/CREB proteins is their role in cell adaptation and maintenance of homeostasis [4,6,7,8]. For example, ATF3, a member of the ATF/CREB family, is an immediate early gene involved in the adaptive cellular response and is upregulated by various intracellular and extracellular signals such as injury and signals related to cell proliferation [6]. ATF3 is also a member of the RAG group of transcription factors [9]. The important role of RAG transcription factors in axonal neurite outgrowth and neuronal regeneration has been assessed and confirmed both in vitro and in vivo, in PNS and CNS neurons, either by pharmacological inhibition or genetic attenuation [5,10,11,12,13]. In addition to ATF3, in vivo axonal growth is also stimulated by other RAG transcription factors such as Smad1 [14], STAT3 [15,16], c-Jun [17], and ATF1/CREB [18].

We have previously demonstrated the involvement of ATF3 in the control of ependymal stem cells activity in the rat neonatal spinal cord and shown that ATF3 can be considered a novel marker for spinal migratory or quiescent ependymal stem cells [19].

Numerous in vitro models have been developed to study complex molecular mechanisms involved in CNS axon regeneration, demonstrating that many aspects of neuronal development are faithfully reproduced in vitro [20,21]. For example, comparative studies of the fetal brain and in vitro neurospheres have shown that in vitro neuronal progenitor cells act as their in vivo correspondents and that the proliferative capacity and potential for cell differentiation coincide with the cell fate and developmental stage of the tissue from which the cells are derived [22,23].

Thus, we have recently established the primary neuronal and radial glial cell (RGC) cultures from neonatal opossums *Monodelphis domestica*, where RGC retain their neurogenic potential, confirming that neuronal differentiation occurs in vitro [24]. Opossums are born very immature, and postnatal day 0 (P0) neonates correspond to E12 rats, while P14 opossums correspond to P0 mice or rats [25]. Opossums have a unique ability to fully regenerate the spinal cord after injury during the first two weeks of their postnatal life, after which that ability abruptly ceases. This unique regenerative ability makes opossums particularly suitable for studying key regulators of CNS regeneration [26,27]. Thus, primary cortical neurons from neonatal opossums offer a valuable in vitro model for studying the mechanisms underlying regeneration. To date, there are no data on the ability of the opossum to regenerate the brain after injury, and this research is the first attempt to establish an in vitro platform for opossum brain regeneration research.

Here we compare the gene expression levels of ATF and RAG family members between opossum neuronal progenitor cells and differentiated neurons and investigate the role of ATF3 in neuronal survival and differentiation, as well as in neurite outgrowth, neuronal network formation, and CNS neuroregeneration.

This is also the first report of long-term primary cortical cultures (>DIV14) from the postnatal mammalian CNS in which regenerative neurite outgrowth occurs after injury.

Investigating changes in gene transcription in response to axonal transection in opossum cortical cells may provide new insights into molecular pathways essential for neuronal differentiation and regeneration.

## 2. Materials and Methods

### Animals

The colony of South American gray short-tailed opossum (*Monodelphis domestica*) is housed at the animal facility of the University of Trieste in accordance with the guidelines of the Italian Animal Welfare Act. Their use has been authorized by the Ethics Committee board, the Local Veterinary Service, and the National Ministry of Health (Permit Number: 1FF80.N.9Q3) in accordance with European Union guidelines for animal care (d.1.116/92, 86/609/C.E.). Animals are kept in standard laboratory cages with controlled temperature (27–28 °C) and humidity (50–60%) with a 12/12 h light/dark cycle and ad libitum access to food and water. The experiments were carried out in accordance with the European Directive 2010/63/EU for animal experiments, and every effort was made to minimize the animal suffering and the number of animals used.

## 3. Primary Cortical Cultures

Primary cortical cultures were prepared as described in the work of (24). Briefly, cells were isolated from the cortex of neonatal opossums of both sexes at postnatal days (P) 5–6 and 16–17. Dissection was performed in ice-cold oxygenated dissection solution (113 mM NaCl, 4.5 mM KCl, 1 mM MgCl_2_ × 6H_2_O, 25 mM NaHCO_3_, 1 mM NaH_2_PO_4_, 2 mM CaCl_2_ × 2H_2_O, 11 mM glucose and 0.5% *w/v* penicillin/streptomycin/amphotericin B; all from Sigma-Aldrich, St. Louis, MO, USA), pH 7.4. The cortices were washed three times with sterile phosphate-buffered saline (PBS, 137 mM NaCl, 2.7 mM KCl, 10 mM Na_2_HPO_4_, 2 mM KH_2_PO_4_; all from Sigma-Aldrich).

Enzyme digestion was performed by incubation in trypsin (0.5% Trypsin-EDTA Solution 10X, cat. no. sc-363354, Santa Cruz Biotechnology, SCBT, Dallas, TX, USA) at 32.5 °C. P5–6 cortices were incubated in 0.5% trypsin in PBS for 10 min, while P16-17 cortices were incubated in 2.5% trypsin for 15 min. Cells were dissociated in a trituration solution containing 10 µg/mL DNAse I (Sigma-Aldrich), 1 mg/mL trypsin inhibitor (SCBT), and 1% bovine serum albumin (BSA, Pan-Biotech GmbH, Aidenbach, Germany) in HBSS solution, w/o Ca^2+^ and Mg^2+^ (Pan-Biotech). The supernatant containing the dissociated cells was collected and deposited on top of a 5% BSA cushion in HBSS in a 5 mL tube, centrifuged for 5 min at 100× *g*, and then resuspended in a plating medium consisting of Dulbecco’s Minimum Essential Medium (DMEM) with stable glutamine supplemented with 10% fetal bovine serum (FBS) and 1% penicillin/streptomycin (all from Pan-Biotech). Prior to plating, the cell suspension was preplated onto a plastic tissue culture Petri dish for 5 min.

For RT q-PCR experiments, cells were plated in a T25 tissue culture flask in a plating medium at the density of 6 × 10^5^ cells. To grow the neurospheres in suspension, the cells were plated in a non-adherent T25 tissue culture flask in DMEM with stable glutamine supplemented with 10% FBS and 1% penicillin/streptomycin used for both plating and culturing. The next day, the cells were transferred to a new tissue culture flask to remove adherent cells. A quarter of the media was changed at DIV1 and DIV4. For neuronal culture, cells were plated in an adherent T25 tissue culture-treated flask coated with 50 µg/mL poly-L-ornithine. The next day, three-quarters of the plating media were removed, and a neuronal medium containing neurobasal medium supplemented with B27 (both from Thermo Fisher, Waltham, MA, USA), 1 mM L-glutamine and 1% penicillin/streptomycin (both from Pan-Biotech) was added. At DIV4, half of the medium was changed with fresh neuronal medium. Cells were maintained at 32 °C, 5% CO_2_, and 95% relative humidity.

### 3.1. Neuroregeneration Scratch Assay

For scratch assay and immunocytochemistry experiments, cells were plated on 12 mm diameter glass coverslips precoated with 50 µg/mL poly-L-ornithine and 2 µg/mL laminin (all from Sigma-Aldrich) at a density of 0.5 × 10^5^ cells per well in a 24-well plate. The next day, two-thirds of the medium was changed with neuronal medium. Half of the medium was replaced with fresh neuronal medium once a week. At DIV10, DIV14 or MIV1, a scratch was made in the middle of the coverslip severing the differentiated neurons with fine-tipped tweezers thick 0.1 mm, resulting in a scratch approximately 100 μm wide. The neurons were immediately imaged with a Zeiss inverted microscope. Cells were fixed and immunostained 24 or 48 h after scratching.

### 3.2. Drug Protocols

JNK/c-Jun inhibitor SP600125 and MAPK/p38 inhibitor SB203580 (both from SCBT, cat.no. sc-200635 and sc-3533, respectively) were introduced into neuronal culture with or without injury, at different DIV (as explained in the Section 8), at a final concentration of 50 and 1 µM, respectively, in PBS and were incubated for 24 h.

### 3.3. Quantitative RT-qPCR

RNA from primary cortical neurons and neurospheres was isolated using a Monarch Total RNA Miniprep kit (New England Biolabs (NEB), Ipswich, MA, USA) according to the manufacturer’s instructions. DNAse I (NEB) was added to the RNA isolation column to eliminate genomic DNA. RNA concentration and purity were analyzed on a Biodrop Duo spectrophotometer (Harvard Bioscience, Holliston, MA, USA), and the quality was then analyzed on a 2% agarose gel (Sigma-Aldrich). To assess the purity of the samples, the absorbance ratio values at 260 nm and 280 nm (A260/A280) and at 260 nm and 230 nm (A260/A230) were considered and taken as good (indicating the high purity of RNA) when the A260/A280 was about 2.2 and the A260/A230 ranged from 1.8 to 2.2.

The QuantiTect RT kit (Qiagen, Hilden, Germany) was used to transcribe RNA into cDNA. A total of 1 μg of total RNA was transcribed into cDNA in a 20 μL reaction buffer. For a single RT-qPCR reaction, 5 μL of cDNA (diluted 10× in dH_2_O) was used as a starting material.

The RT-qPCR reaction was performed with Luna Universal qPCR Master Mix (NEB) based on SYBR Green I dye using LightCycler 480 (Roche Holding AG, Basel, Switzerland). The RT-qPCR program consisted of preincubation for 10 min at 95 °C, 50 cycles of DNA amplification with the following steps: 10 s at 95 °C, 10 s at 55 °C, and 20 s at 72 °C, and final melting and cooling. Glyceraldehyde-3-phosphate dehydrogenase (GAPDH) was used as a reference gene. The cycle threshold (Ct) of the GAPDH gene ranged from 14 to 16 cycles. The cut-off value for Ct was 35. Non-template controls were performed for each primer pair used in the RT-qPCR reaction. The relative mRNA expression levels of all genes of interest (GOI) were normalized to the level of the reference gene (GAPDH) and calculated by the equation 2^ (Ct value for GAPDH—Ct value for GOI). All samples were made in technical duplicate and biological triplicate.

The primers for genes of interest were designed using the Primer3Plus (https://primer3plus.com/, accessed on 1 April 2020) web interface [28], and transcripts for the opossum were downloaded from the Ensembl database (https://www.ensembl.org, accessed on 1 April 2020). The sequence specificity of designed primers was confirmed using the NCBI Nucleotide BLAST tool (https://blast.ncbi.nlm.nih.gov/Blast.cgi, accessed on 1 April 2020). The synthesis service of selected primers was performed by Macrogen (Macrogen Europe, Amsterdam, Netherlands). The initial PCR amplification products were run on a 2% agarose gel to verify that the primer pairs multiplied a single product of the predicted size. The change in gene expression of interest is expressed as a “fold change” (^2^(-ΔΔCt)). The mRNA level in the control sample was designated as 1. Scatter plots of relative mRNA expression normalized to the mRNA level of the GAPDH reference gene (ΔCt) with statistical tests performed and *p*-values are shown in Appendix A.

The primers used were: *cJUN* forward 5′-CAAGTGCCGGAAAAGGAAGC-3′, reverse 5′-CGCTGTTCACGTGGTTCATG-3′; *STAT3* forward 5′-TGCAGCATTAAGAGGATCCCG-3′, reverse 5′-GAAGCATCACAATTGGCCCG-3′; *SMAD1* forward 5′-TTCCAGATGCCAGCGGATAC-3′, reverse 5′-AACTGCCTGCACATCTCCTC-3′; *ATF1* forward 5′-TCAGAGACAGCACCACAACC-3′, reverse 5′-AATCCCCCGAGCTTTCTGTG-3′; *ATF2* forward 5′-ACACCTACACCAACACGATTCT-3′, reverse 5′-TGATGGGTGTTGCAAGAGGG-3′; *ATF3* forward 5′-AGTTTGCCCCTGAAGAGGATG-3′, reverse 5′-CCAACTTTTCTGATTCCTTCTGC-3′; *ATF4* forward 5′-ACAGACTTTGGCAAGGAGGATG-3′, reverse 5′-ATCACAAGAGCCTTCCAACG-3′; *ATF6* forward 5′-ACCAGCATCAGGAATTCAGGG-3′, reverse 5′-AATAGCAGGTGATCCCGTCG-3′; *ATF7IP* forward 5′-GAGTTGAGAACCAGACCAGCA-3′, reverse 5′-ACACCTCCTGAATCACTGCC-3′; *GAPDH* forward 5′-ATGCCCCAATGTTCGTGATG-3′, reverse 5′-GTCATGAGTCCTTCCACAATGC-3′.

## 4. Immunofluorescence Staining

Cells were fixed for 20 min at room temperature (RT, 20–22 °C) with 4% paraformaldehyde (PFA) pH 6.9 containing 200 mM sucrose (all from Sigma-Aldrich) in PBS. After fixation, cells were washed with PBS, saturated with 0.1 M glycine, permeabilized with 0.1% Triton X-100 (all from Sigma-Aldrich) in PBS, and finally washed with PBS, each step for 5 min. Cells were then incubated with 0.5% BSA (Pan-Biotech) in PBS blocking solution for 30 min.

Incubation with primary antibodies was performed in a humid chamber for 2 h, followed by 2 washing steps (5 min each) in PBS. The primary antibodies were following: ATF3 (Abcam, ab216569, 1:50, Uniprot Align immunogen identity: 92.3%; antibody validation shown in Appendix A), SOX2 (Abcam, ab79351, RRID: AB_10710406, 1:200, 91.7%), β-tubulin III (TUJ1; Biolegend, 801201, RRID: AB_2313773, 1:200, 99.8%). Additional information on primary antibodies can be found in Appendix A.

Cells were incubated with secondary antibodies containing a 300 nM nuclear stain 4′, 6-diamidino-2-phenylindole (DAPI, Thermo Fisher Scientific, Waltham, MA, USA) in PBS. The incubation time was 1 h in a dark, humid chamber. The secondary antibodies were following: goat anti-mouse Alexa Fluor^®^ 488 (Thermo Fisher Scientific, A32723, RRID: AB_2633275, 1:400), goat anti-rabbit Alexa Fluor^®^ 555 (Thermo Fisher Scientific, A32732, RRID: AB_2633281, 1:400), goat anti-rabbit Alexa Fluor^®^ 647 (Abcam, ab150083, RRID: AB_2714032, 1:300), goat anti-mouse IgG1 Alexa Fluor^®^ 488 (Thermo Fisher Scientific, A-21121, RRID: AB_2535764, 1:300) and goat anti-mouse IgG2a Alexa Fluor^®^ 555 (Thermo Fisher Scientific, A-21137, RRID: AB_2535776, 1:300).

Next, the coverslips were washed twice with PBS, once in dH_2_O, mounted on a glass slide with a mounting medium (Vectashield, Vector Laboratories, Burlingame, CA, USA), and sealed with nail polish. All incubations were performed at RT.

## 5. Imaging

Samples were analyzed using an Olympus IX83 inverted fluorescent microscope (Olympus, Tokyo, Japan) equipped with differential interference contrast (DIC) and fluorescence optics (mirror units: U-FUNA: EX360-370, DM410, EM420-460, U-FBW: EX460-495, DM505, EM510IF, and U-FGW: EX530-550, DM570, EM575IF, all from Olympus and Cy5: EX620/60, DM660, EM700/75, Chroma, Bellows Falls, VT, USA).

Fluorescence images were acquired with Hamamatsu Orca R2 CCD camera (Hamamatsu Photonics, Hamamatsu, Japan) and CellSens software (Olympus, Japan), with slice spacing of 1.27 µm for 20× objective (0.5 NA) and 0.29 µm for 40× oil immersion objective (1.4 NA). For each image, an average intensity projection was used. CellSens and ImageJ (http://rsbweb.nih.gov/ij, accessed on 1 April 2020) (NIH, Bethesda, Maryland, USA) were used for image processing and analysis.

ImageJ SNT plugin was used to measure the average neurite length. ImageJ was used to quantify immunofluorescence signals (gray level value expressed in arbitrary units, AU) by measuring the average TUJ1 pixel intensity in the 100 × 100 μm region of interest (ROI). The average background pixel value was measured for each image and subtracted. TUJ1 gray values are quantified in the scratch area and normalized with TUJ1 averaged gray values of intact neuronal networks. CellSens software was used for line scan analysis of the nuclear ATF3 immunofluorescent signal. To obtain a line plot, an analysis was performed on the average intensity projection of the Z-stack. Neuronal culture samples were immunoprocessed in parallel under identical conditions and imaged using the same exposure settings.

Cell survival was assessed by comparing the number of DAPI-positive intact and pyknotic nuclei between control conditions (SHAM) and MAPK inhibitors or antibody-treated cultures.

## 6. Electrophoresis and Immunoblotting

To detach the neurons from the substrate, cells were washed with PBS and incubated with 0.5% trypsin for 5 min at 32 °C. DMEM supplemented with 10% FBS was added to inactivate trypsin. The cells were collected in pre-cooled 15 mL tubes and centrifuged at 300× *g* for 5 min at 4 °C. The supernatant was removed, and the pellet was briefly washed with ice-cold PBS.

Cytoplasmic and nuclear fractions were isolated according to the Nuclear Extract Kit Assay Protocol (Abcam, Cambridge, UK). First, 250 µL of ice-cold complete hypotonic buffer (1×) was added, and the cells were incubated on ice for 15 min. Then 50 µL of 10% NP-40 assay reagent was added to each sample and mixed gently by pipetting. The samples were centrifuged at 14,000× *g* for 30 s at +4 °C in a tabletop microcentrifuge. The supernatant with the cytosolic fraction was transferred to a new 1.5 mL tube. The pellet was resuspended in 50 µL ice-cold complete nuclear extraction buffer, and tubes were vortexed vigorously for 30 s and then gently rocked at +4 °C for 15 min and repeated once more. The samples were centrifuged at 14,000× *g* for 10 min at 4 °C, and the supernatant with the nuclear fraction was set aside.

Protein lysates were mixed with 4× Laemmli buffer (50 mM Tris pH 6.8, 10% glycerol, 2% SDS, DTT, and 0.04% bromophenol blue, all from Sigma-Aldrich) to a final volume of 25 µL and denatured on a thermoblock for 5 min at 95 °C.

Electrophoresis was performed at 90 V for a 4% stacking gel, followed by 150 V for a 12% sodium dodecyl sulfate (SDS) polyacrylamide separation gel. Proteins were transferred to a nitrocellulose membrane (GE Healthcare, Chicago, IL, USA) for 1 h at 100 V. Membranes were blocked in a 5% solution of bovine serum albumin (BSA, Pan-Biotech GmbH, Aidenbach, Germany) in TBS-T (0.1% Tween 20 in Tris-buffered saline, pH 7.4, all from Sigma-Aldrich), one hour on a shaker at RT. After blocking, the membranes were incubated with the primary ATF-3 antibody solution in 3% BSA in TBS-T (SCBT, sc-188, RRID: AB_2258513, 1:100; antibody validation shown in Appendix A) overnight on a shaker at +4 °C.

The next day, the membranes were washed 3× for 5 min in TBS-T and then incubated in the secondary antibody solution for 1 h at RT on a shaker. The secondary antibodies were horseradish peroxidase (HRP)-linked goat anti-rabbit (7074, RRID:AB_2099233, 1:2000) and goat anti-mouse (7076, RRID: AB_330924, 1:2000; both from Cell Signaling Technology, CST, Danvers, MA, USA) and were diluted in 1% BSA in TBS-T. After incubation, the membranes were washed 3× for 5 min in TBS-T. After washing, the chemiluminescent reagent Liteablot Turbo (Euroclone Diagnostica, Milan, Italy) was applied to the membrane and incubated for 1 min at RT. Results were acquired using the ChemiDoc MP Imaging System (Bio-Rad, Hercules, CA, USA).

GAPDH (Proteintech, HRP-60004, AB_2737588, Rosemont, IL, USA) was used as the loading control for the protein levels in the cytoplasmic fractions, and Histone H4 (CST, 2935, RRID: AB_1147658) was used as the loading control for the protein levels in the nuclear fractions. Both loading controls were prepared in 3% milk powder in TBS-T at a dilution of 1:5000.

## 7. Data Analysis

Statistical analysis was performed using GraphPad Prism 8.4 (GraphPad Software Inc., San Diego, CA, USA). D’Agostino–Pearson or Shapiro–Wilk normality test was used to calculate how far the values differ from the values expected with a Gaussian distribution. The normality test was chosen depending on the number of values tested. The Brown-Forsythe test was used to test the assumption that all data are sampled from populations with the same standard deviations.

Three or more data groups were compared using one-way ANOVA, based on the assumption that the populations are Gaussian and that the variances are equal. Following the one-way ANOVA parametric test, the post-hoc Holm-Šídák test was used for multiple comparisons between the two data groups. Data groups with different variances that failed a normality test were compared using the Kruskal–Wallis test. Following Kruskal–Wallis nonparametric test, post-hoc Dunn’s multiple comparisons test was used. Data groups with equal variances assumed to be sampled from a Gaussian distribution were compared using the Welch ANOVA and post-hoc Dunnett’s T3 multiple comparisons test (for *n* < 50). The *p*-value was adjusted for multiple comparisons.

An unpaired *t*-test was used when comparing two normally distributed data groups. The *t*-test assumes that the two samples come from populations that have identical standard deviations was tested using the F test. In case of unequal variances, Welch’s *t*-test was used. The accepted level of significance was *p* < 0.05. *p* < 0.001 Very significant ***, 0.001 to 0.01 Very significant **, 0.01 to 0.05 Significant *, ≥0.05 Not significant. The decimal format used to report *p*-values was according to the recommended NEJM (New England Journal of Medicine) style (https://www.graphpad.com/guides/prism/latest/statistics/stat_decimal_formatting_of_p_values.htm, accessed on 20 January 2021; https://www.graphpad.com/guides/prism/latest/statistics/extremely_significant_results.htm, accessed on 20 January 2021).

## 8. Results

### 8.1. The Significant Difference in RAG and ATF Gene Expression in Opossum’s Cortical Differentiated Neurons versus Neurospheres

We investigated the expression of RAG and ATF/CREB genes by RT-qPCR in primary cortical cultures obtained from P5-6 and P16-17 opossums because these particular age groups differ substantially in their neuroregenerative capacity after spinal cord injury [26,29,30,31,32]. Primary cultures were prepared using different culture conditions to obtain differentiated neurons or undifferentiated neurospheres (see Materials and Methods and [24]). Neurospheres mainly consist of Paired Box Gene 2 (PAX2; [33]) and SRY-Box Transcription Factor 2 (SOX2; [34])-positive neuronal progenitors (Appendix A; [24]).

As shown in Figure 1, after 1 week in culture, significantly different RAG and ATF gene expressions were detected when comparing neurospheres and neurons, regardless of whether the cultures were prepared from younger (P5-6) or older animals (P16-17). We observed significantly higher expression of RAGs: c-Jun, STAT3, and SMAD1 (Figure 1A–C), and ATFs: ATF1, ATF2, ATF3, ATF4, and ATF6 (Figure 1D–H) in neurospheres compared to neurons. The highest difference was observed for ATF3 (Figure 1F), with a more than eight-fold higher expression in neurospheres compared to differentiated neurons.

Interestingly, we observed significantly higher levels of SMAD1, ATF1, ATF2, and ATF3 expression (Figure 1C–F) in P6 neurons compared to P17 neurons. Similarly, P6 neurospheres had higher levels of c-Jun, STAT3, ATF2, ATF3, and ATF6 (Figure 1A,B,E,F,H) compared to P17 neurospheres. Finally, no significant difference in ATF7 mRNA levels was found in any comparison (Figure 1I).

### 8.2. ATF3 Cytoplasmic Translocation Occurs during Neuronal Differentiation

We investigated cellular localization of the ATF3 in opossum cortical undifferentiated neurospheres (Figure 2A) and in neurospheres induced to differentiate for 4 days. Figure 2B shows the transition from neuronal progenitors (Figure 2A) to a differentiated neuronal phenotype, as indicated by the β-tubulin III-positivity [35,36] and extensive neurite outgrowth.

A strong ATF3 fluorescence signal was detected in both primary cultures; however, the staining pattern was different: in neuronal progenitors, staining was nuclear (Figure 2A), while in differentiated neurons, staining was predominantly cytoplasmic, with a low signal in the nucleus (Figure 2B), which was confirmed by the line scan analysis of ATF3 fluorescence intensity (Figure 2C,D). Substantially higher ATF3 expression in neuronal progenitors is consistent with the ATF3 levels measured by RT-qPCR (Figure 1F).

We also performed Western blot analysis using nuclear and cytoplasmic fractions from neurospheres and neurons (Figure 2E). We detected a 20 kDa ATF3 isoform abundant in the nuclear fraction of neurospheres, which was not present in the neuronal nuclear fraction.

### 8.3. Role of ATF3 in Neurite Outgrowth and Cell Survival

Previous studies demonstrated that the JNK [37,38,39] and the p38 [40] signaling pathways are involved in the induction of ATF3 transcription.

Therefore, to investigate whether ATF3 regulates neurite outgrowth, neuronal network formation, or cell survival, we pharmacologically inhibited ATF3 in P5 and P17 primary cortical neuronal cultures with MAPK inhibitors: JNK/c-Jun inhibitor SP600125 (SP) or p38-MAPK inhibitor SB203580 (SB).

Inhibitors were added either at DIV1 (Figure 3) when cultures were mainly composed of immature neurons, positive for TUJ1, SOX2, and brain lipid-binding protein (BLBP; Appendix A; [41]), or at DIV4 where neuronal networks were prominent (Figure 4).

In both P5 and P17 neuronal cultures, SP (Figure 3A2,B2) and SB inhibitor (Figure 3A3,B3), applied at DIV1, prevented the neurite outgrowth and elongation and neuronal network formation. As a result, significantly shorter neurites were present in both P5 (Figure 3C1) and P17 neuronal cultures (Figure 3C2). The average neurite length of 100 μm in control conditions was shortened to only 35 μm after administration of SP inhibitor or to only about 40 μm after administration of SB inhibitor.

Moreover, both inhibitors adversely affected cell viability. The SP inhibitor caused a 25% increase in cell death in P5 (Figure 3D1) and a 30% increase in P17 cultures (Figure 3D2), while the SB inhibitor caused a 23.5% increase in cell death in P5 (Figure 3D1) and 17.5% in P17 cultures (Figure 3D2), compared to control conditions.

When applied at DIV4, thus when cultures consist of differentiated neurons forming dense networks, neither of the two inhibitors affected the length of neurites in P5 or P17 cultures (Figure 4C1,C2), nor there was a significant increase in cell death (Figure 4D1,D2). However, the perinuclear aggregation of ATF3 was observed in the cytoplasm of neurons after administration of inhibitors, and the fluorescence intensity of ATF3 in neuronal nuclei decreased by more than 50% after administration of SP or SB inhibitors (Figure 4E1,E2 and Appendix A).

### 8.4. Development of the Neuroregeneration Scratch Test to Measure Neurite Outgrowth after Mechanical Injury in Neonatal Opossum Cortical Cultures

Previous studies demonstrated different abilities of the P5 and P17 opossum’s spinal cords maintained in vitro to regenerate after injury [26,30]. Here, we tested the regenerative capacity of P5 and P17 opossum primary cortical neurons using a scratch test [21] that allows relatively easy and rapid measurement of neurite outgrowth after mechanical injury. To confirm the complete removal of cell bodies and neurites from the scratched area, we imaged live cells immediately after scratch (Figure 5A) and immunostained them for TUJ1 (Figure 5B).

To investigate the regenerative ability of neurons, a scratch was made in both P5 and P17 cultures (Figure 5C1,D1). Neurite outgrowth and migration of immature neurons (TUJ1+/SOX2+) into the scratched area were observed 24 h after injury. Furthermore, 48 h after injury, the number of neurites crossing the scratched area increased significantly (Figure 5C2,D2), indicating a determined regeneration progression. This is similar to the response of rat cortical neurons in vitro that begin to regenerate after a few hours following axonal injury, in contrast to sensory neurons that typically start axonal regeneration almost immediately [42].

Remarkably, we observed that neurons derived from young opossums (P3) successfully survive and establish neuronal networks capable of regeneration even after 1 month in vitro (Figure 5E), also showing immature neurons migrating to the injury site.

### 8.5. The Role of ATF3 in Neuronal Regeneration after Injury

To test the regenerative ability of P5 and P16 differentiated neurons, a scratch test was performed after two weeks in vitro, and SP or SB inhibitors were used to investigate the role of ATF3 in neurite outgrowth and regeneration after injury.

Under control conditions, 24 h after the injury, growth cones (Appendix A) and neurites crossing the scratched area were visible (Figure 6A1,B1). In contrast, retraction bulb formation was predominant in cultures after ATF3 inhibition (Figure 6A2,A3,B2,B3).

In P5 cultures, after SP or SB inhibitor administration (Figure 6C1), TUJ1 pixel density in the scratched area decreased similarly, on average three-fold (from 0.6 to 0.2 AU). Furthermore, the average neurite length of P5 neurons in the control condition was >115 μm, while inhibitor-treated neurons had significantly shorter neurites, 75 μm and 79 μm, respectively (Figure 6D1).

In the P16 neuronal cultures, the TUJ1 pixel density in the scratched area decreased almost two-fold, from 0.75 to 0.4 AU, similarly after SP or SB application (Figure 6C2). Additionally, in P16 control cultures, the average neurite length was >105 μm, compared to SP- or SB-treated cultures, where the length significantly decreased to 66 μm and 74 μm, respectively (Figure 6D2). Thus, the inhibitors had a similar effect on injured neurons derived from P5 or P16 opossums, with a similar decrease in neurite density and length (Figure 6C1,C2,D1,D2). No significant increase in cell death was observed in P5 or P16 cultures following SP or SB inhibitor application after injury. The average cell death was 25–30% (Figure 6E1,E2). The fluorescence intensity of ATF3 in neuronal nuclei decreased by more than 50% (Figure 6F1–G2; Appendix A) after the administration of inhibitors.

Next, we tested the effect of the prolonged ATF3 inhibition on neurite outgrowth and regeneration of opossum cortical neurons in vitro. Therefore, when the SP inhibitor was administered 24 h before the cut (Figure 7A2,B2), the TUJ1 pixel density decreased almost two-fold: in P5 cultures from 0.68 to 0.31 AU (Figure 7C) and in P16 cultures from 0.63 to 0.35 AU (Figure 7D), compared to control conditions.

To test whether neurons would recover and regenerate after inhibitor removal (Figure 7A4,B4), the SP inhibitor was administered immediately after the cut for the next 24 h, followed by a 24 h washout (Figure 7E). The TUJ1 pixel density decreased almost two-fold: in P5 cultures from 0.75 to 0.48 AU (Figure 7C) and in P16 cultures from 0.67 to 0.31 AU (Figure 7D).

The growth of new neurites was visibly reduced after treatment with the SP inhibitor, whether the inhibitor was administered before or after injury, indicating a prolonged effect of ATF3 inhibition on neurite outgrowth and neuronal regeneration (Figure 7A1,A3,B1,B3).

Finally, to verify if the effect of the JNK/c-Jun and p38-MAPK inhibitors on neuronal growth and regeneration after injury was, in fact, associated with ATF3 and not with other potential molecular factors, we blocked ATF3 using its specific antibody (Figure 8).

Thus, the ATF3 antibody was applied to P5 and P16 neuronal cultures immediately after the scratch for the next 24 h, after which the neurite length, TUJ1 pixel density, and cell death were quantified. In the P5 cultures, the average TUJ1 pixel density decreased from 0.65 ± 0.15 to 0.25 ± 0.08 AU, (Figure 8C), and similarly in P16 cultures decreased from 0.65 ± 0.12 to 0.37 ± 0.09 AU (Figure 8C). The average neurite length in P5 cultures decreased from 100 to 70 μm (Figure 8D), while in P16 cultures decreased from 97 to 72 μm (Figure 8D) compared to the control condition.

Importantly, the percentage of dead cells was significantly higher after administration of ATF3 antibody in both P5 (10% increase) and P16 cultures (25% increase) compared to control conditions (Figure 8E).

Thus, the administration of ATF3 antibody immediately after injury in the opossum neuronal cultures significantly reduced the average length and density of growing neurites in the scratch area, with a significant increase in cell death.

## 9. Discussion

### 9.1. The Different Expression and Localization of ATF3 in Neural Progenitor Cells versus Opossum Cortical Neurons

ATF3, which we previously found to be important in the control of activity and migration of endogenous spinal stem cells [19], was now investigated as a possible regulator of neuronal differentiation, network formation, and regeneration of primary cortical neurons derived from postnatal opossums, that we have established recently [24]. These primary cultures showed remarkably similar time course of in vitro network formation and maturation when compared to the well-established cortical or hippocampal rodent cultures [43,44,45,46], such as the formation of neuronal growth cones at DIV1, formation of axons and MAP2-positive dendrites from DIV2-3 onwards, followed by the expression of synaptic markers such as synapsin (Appendix A) during the first two weeks of the in vitro maturation. Moreover, the nearly pure neuronal cultures obtained using P5-6 opossums (that corresponds to late rodent embryos) and the enrichment in glial cells in cultures prepared from P16-17 cortex (similar to rodent neonates) correlates well with cultures derived from mice or rat brain of equivalent age [25,47].

Thus, primary cultures of cortical neurons and neurospheres were prepared from opossums at different postnatal ages with different spinal cord regeneration potential, and the difference in RAG and ATF gene expression was investigated. We found significantly higher levels of RAG and ATF members in neuronal progenitors compared to neurons. Likewise, the cultures derived from younger (P6) opossums had higher expression of RAG and ATF/CREB genes when compared to P17 cortical cultures, suggesting their importance in the CNS development of neonatal opossums. Among those, the increase in ATF3 mRNA was the strongest and clearest. Moreover, we detected the different localization of the ATF3 in neuronal progenitors (nuclear) and differentiated neurons (predominantly cytoplasmic), suggesting that the ATF3 translocation occurs during neuronal differentiation. This correlates with the previously demonstrated nuclear translocation of ATF3 in active migrating ependymal cells in rat spinal cord, as opposed to cytoplasmic ATF3 localization in quiescent ependymal cells [19].

Furthermore, we detected the 20 kDa ATF3 isoform abundant in the nucleus of neuronal progenitors, which was not detected in the nucleus of differentiated neurons, indicating the existence and different subcellular localization of ATF3 isoforms in neuroprogenitors and neurons. There are five alternatively spliced isoforms of ATF3 known in humans: ATF3 full-length, ATF3-Δzip, ATF3-Δzip2 (ATF3Δzip2a and ATF3Δzip2b), ATF3-Δzip2c, ATF3-Δzip3, and ATF3b, while only two predicted opossum ATF3 transcript variants are currently available in the Ensembl database: full 20 kDa and short 14 kDa. The alternatively spliced 14 kDa ATF3 isoform is thought to correspond to human ATF3 ΔZip, since it does not contain a leucine zipper domain or bind DNA, and it separates inhibitory cofactors from the promoter initiating transcription, unlike the full ATF3 isoform [48].

The diverse roles of ATF3 in different tissues can also be explained by the posttranslational modifications that the protein undergoes [49]. Consistent with this idea, the detected ATF3 bands around 30 and 35 kDa (Figure 2E) are most likely sumoylated ATF3 full-length isoforms because the sumoylation is the most common ATF3 posttranslational modification [49].

### 9.2. The Role of ATF3 in Neurite Outgrowth, Neuronal Differentiation, and Network Formation during Postnatal Development

To demonstrate the role of ATF3 in neuronal differentiation, neurons were incubated with JNK/c-Jun (SP) or MAPK/p38 (SB) inhibitors. Both inhibitors prevented the neurite outgrowth and the neuronal network formation when applied during the initial stages of neuronal growth (i.e., DIV1), indicating a key role of ATF3 in neuronal differentiation and development. Other studies supporting this hypothesis showed that SP reversibly blocks the development of neuronal polarity and axon formation [50] and that JNK1-deficient embryonic stem cells do not differentiate into neurons [37]. It is interesting that if SP and SB inhibitors were added later to the opossum cortical neuronal cultures after the formation of neuronal networks, no significant cell death or detrimental effect on neurite outgrowth was detected. This is consistent with Tiwari et al., who observed that inhibition of JNK during early neurogenesis causes cell death and termination of neurite growth, whereas inhibition of JNK after neuronal differentiation does not cause cell death [51].

However, we did observe perinuclear aggregation of ATF3 in the cytoplasm of differentiated neurons following SP or SB inhibitor administration. This suggested possible inhibition of ATF3 translocation in the nucleus, as previously demonstrated in the rat spinal stem cells after administration of the inhibitors [19]. Moreover, perinuclear ATF3 aggregates [52] may indicate misfolded proteins, which form insoluble inclusions in the cytoplasm [53].

Our results indicate the important role of ATF3 during early neurogenesis, with the impact on neurite outgrowth and network formation.

### 9.3. Novel Regeneration Scratch Assay

To test the regenerative capacity of P5 and P17 primary opossum cortical neurons, a scratch was made at DIV10 and at DIV15, when neuronal networks can be considered mature [54]. We previously confirmed the existence of excitatory and inhibitory synapses at DIV15 [24]. In both cultures (derived from P5 and P17 opossums), 24 h after injury, we observed a successful growth cone formation, robust outgrowth of neurites, and migration of immature neurons to the injured area, presumably to repair the “wound”. The observed axonal growth rate and cell migration were similar to those seen during embryonic development [55].

There are several studies performed on E18 mouse or rat embryos that investigated the ability of CNS neurons (cortical or hippocampal) to regenerate after transection or axotomy at DIV4 to DIV21 [42,56,57,58]. The successful regeneration was observed only during the early days in vitro, with a later decline in the regeneration ability during neuronal in vitro maturation. Our result contradicts this observation because the opossum cortical neurons successfully regenerated even after 1 month in vitro.

We established and described a new in vitro injury model with postnatal neurons that have an intrinsic ability to regenerate. We cannot, however, exclude the contribution of glial cells that are known to play an important role in the regeneration after injury [59,60]. Further studies are required to address this point, in particular using cultures derived from P16-17 opossums that are enriched with non-neuronal cells (i.e., astrocytes and microglia). In these primary opossum neuronal cultures, various drugs that promote regeneration can be tested, and genes can either be silenced or overexpressed to better understand the molecular basis of CNS regeneration.

### 9.4. The Role of ATF3 in Neuronal Regeneration

ATF3 is considered to play an important role in neurite outgrowth and regeneration of PNS neurons [5,9,11,61], where ATF3 is a marker of active regeneration [62]. Previous studies on DRG showed the importance of ATF3 in axonal outgrowth after injury. Lindwall et al. showed that inhibition of JNK reduces c-Jun phosphorylation and ATF3 expression and decreases neurite outgrowth without affecting neuronal survival [63]. Gey et al. demonstrated reduced RAG response and neurite outgrowth in primary DRG neurons of ATF3 mutant mice, confirming the pro-regenerative role of ATF3 in PNS (11).

However, the involvement of ATF3 in the regeneration of CNS neurons is less understood and more complex. ATF3 has been shown to be neuroprotective after excitotoxic and ischemic brain damage [64]. Furthermore, ATF3 expression is briefly enhanced in damaged CNS neurons and those regenerating into the peripheral nerve graft, where only the prolonged ATF3 expression is associated with successful axon regeneration [65].

The regenerative response is often triggered by retrograde signals coming from the injured axon to neuronal soma [12]. STAT3 and JNK are transported retrogradely as key messengers of injury signaling [66], and Ohara et al. showed that the retrograde transport of STAT3 in injured axons plays a key role in the axonogenesis of hippocampal neurons [57]. Retrograde transport of ATF/CREB members, i.e., ATF2, ATF3, and ATF4, from the injured axons has been observed [67,68], and it is, therefore, possible that ATF3 has a similar function as ATF4 in “sensing the injury” by sending retrograde signals to the neuronal nucleus.

We assume that the localization of ATF3 in the nucleus keeps cells in a multipotent progenitor state and that this potential is attenuated after differentiation. Following injury, ATF3 is transferred to the nucleus to trigger a gene expression pattern similar to that occurring in neuronal progenitors to allow neurite outgrowth, influencing the ability to differentiate and regenerate.

We tested this hypothesis by pharmacological and antibody-mediated inhibition of ATF3, which led to the inability of neuronal progenitors to grow neurites and to differentiate and led to the failure of differentiated neurons to regenerate after injury. Moreover, we observed the formation of aggregates in neuronal cytoplasm after ATF3 inhibition, indicating the inability of ATF3 to translocate into the nucleus.

In conclusion, we have shown the different expression and distinct cellular localization of ATF3 in neural progenitors and differentiated neurons and the involvement of ATF3 in neuronal differentiation, survival, network formation, and regeneration in postnatal opossum cortical neurons using newly established long-term opossum primary cortical cultures and novel neuroregeneration assay.

## Figures and Tables

**Figure 1 ijms-23-04964-f001:**
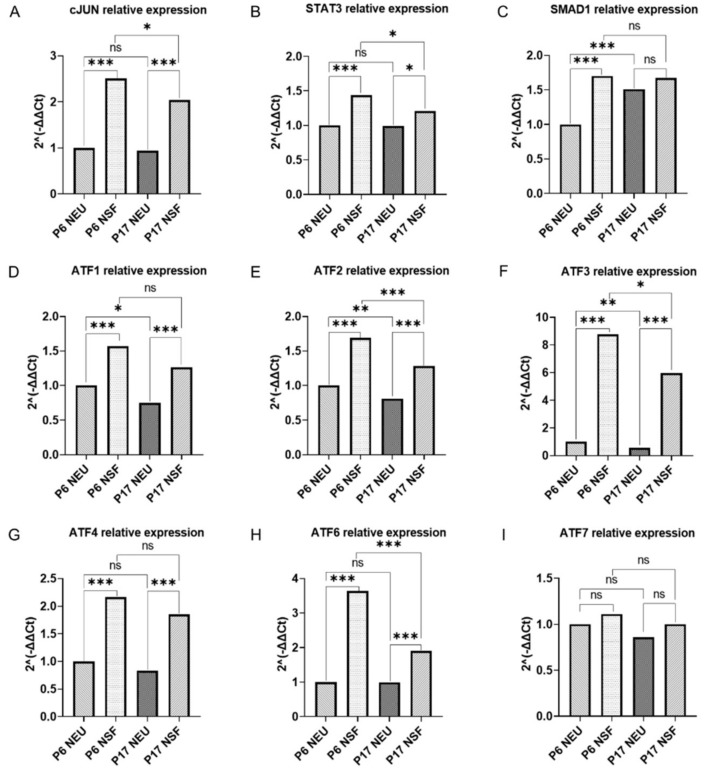
Difference in RAG and ATF gene expression in neurons versus neurospheres. Difference in gene expression between P6 and P17 neurons (NEU) and neurospheres (NSF) was measured by RT-qPCR, normalized to GAPDH mRNA levels, and shown as fold difference compared to P5–6 neurons mRNA levels. mRNA was obtained from cortical neurons, and neurospheres at DIV7 derived from opossums P5-6 and P16-17 from at least three independent experiments. Regeneration-associated genes (**A**–**C**), activating transcription factors (**D**–**I**). The accepted level of significance was *p* < 0.05. *p* < 0.001 very significant ***, 0.001–0.01 very significant **, 0.01–0.05 significant *, ≥0.05 not significant (ns). For statistical significance and tests performed, see Appendix A.

**Figure 2 ijms-23-04964-f002:**
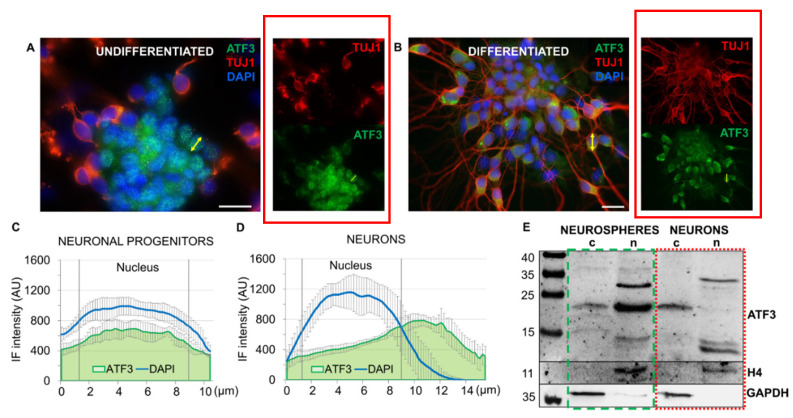
Cellular localization of ATF3 in undifferentiated and differentiated neurospheres. (**A**,**B**) Immunostaining of primary cortical neurospheres obtained from P5-6 opossum cortex: (**A**) undifferentiated neurospheres were cultured in proliferating medium (DMEM/10% FBS) on poly-L-ornithine and fixed at DIV2. (**B**) To induce differentiation, neurospheres cultured in proliferating medium in suspension at DIV3 were transferred on a laminin-coated coverslip, and the medium was switched to neuronal medium for the next 4 days. Cells were fixed at DIV7, and immunostained for ATF3 (green), neuronal cytoskeletal marker β-tubulin III (TUJ1, red), and cell nuclei were counterstained with DAPI (blue). (**C**,**D**) Line scan analysis of primary neuronal progenitors (**C**) and neurons (**D**). Yellow arrows indicate examples of line scan analysis performed on a Z-stack average intensity projection to obtain the plots in which ATF3 (green) and DAPI (blue) are shown in the cytoplasm and nucleus (nuclear region is indicated by the vertical lines). Line plot is the average line intensity of five randomly selected cells. Data are shown as mean ± SD. Scale bar is 20 μm. (**E**) Western blot analysis of ATF3 levels in the nucleus (N) and cytoplasm (C). The loading controls were GAPDH for the cytoplasmic fractions and Histone H4 for the nuclear fractions.

**Figure 3 ijms-23-04964-f003:**
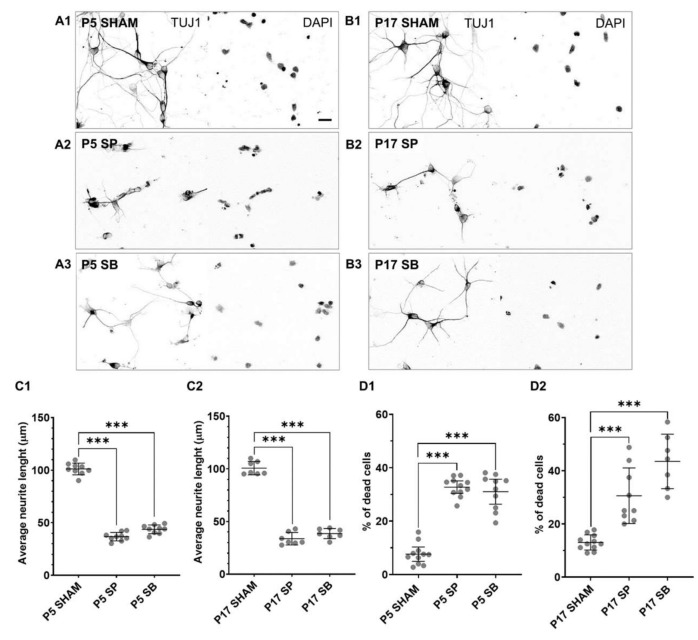
Reticence of neuronal differentiation by ATF3 inhibition. Primary neuronal cultures derived from the P5 (**A1**–**A3**) and P17 (**B1**–**B3**) cortex of *M. domestica* were treated at DIV1 with the MAPK inhibitors, JNK/c-Jun inhibitor SP600125 (SP), or the p38 inhibitor SB203580 (SB) for 24 h. The cells were fixed and stained at DIV7. Neurons were stained for TUJ1 (left panel), and their nuclei were counterstained with DAPI (right panel). For better visualization of neurites, images are shown in the grayscale. Scale bar is 20 μm. (**C1**) The scatter plot shows the average neurite length for P5 cortical neurons for the experimental conditions shown from (**A1**–**A3**). Data are shown as a scatter plot with mean ± SD. The total number of neurites analyzed P5 SHAM 369; SP 272; SB 346. One-way ANOVA followed by Holm-Šídák multiple comparisons test. P5 SHAM vs. P5 SP *p* < 0.001 ***; P5 SHAM vs. P5 SB *p* < 0.001 ***. (**C2**) The scatter plot shows the average neurite length for P17 cortical neurons for the experimental conditions shown from B1-B3. Data are shown as a scatter plot with mean ± SD. The total number of neurites analyzed P17 SHAM 385; SP 181; SB 394. One-way ANOVA followed by Holm-Šídák multiple comparisons test. P17 SHAM vs. P17 SP *p* < 0.001 ***; P17 SHAM vs. P17 SB *p* < 0.001 ***. (**D1**) The scatter plot shows the percentage of dead cells with mean ± SD for the experimental conditions shown from (**A1**–**A3**): The total number of cells analyzed, respectively: 897, 468, 497. One-way ANOVA followed by Holm-Šídák multiple comparisons test. P5 SHAM vs. P5 SP *p* < 0.001 ***; P5 SHAM vs. P5 SB *p* < 0.001 ***. (**D2**) The scatter plot shows the percentage of dead cells with mean ± SD for the experimental conditions shown from (**B1**–**B3**): The total number of cells analyzed, respectively: 437, 164, 185. One-way ANOVA followed by Holm-Šídák multiple comparisons test. P17 SHAM vs. P17 SP *p* < 0.001 ***; P17 SHAM vs. P17 SB *p* < 0.001 ***. For each condition, images of at least three random fields per sample from 3 culture preparations were analyzed.

**Figure 4 ijms-23-04964-f004:**
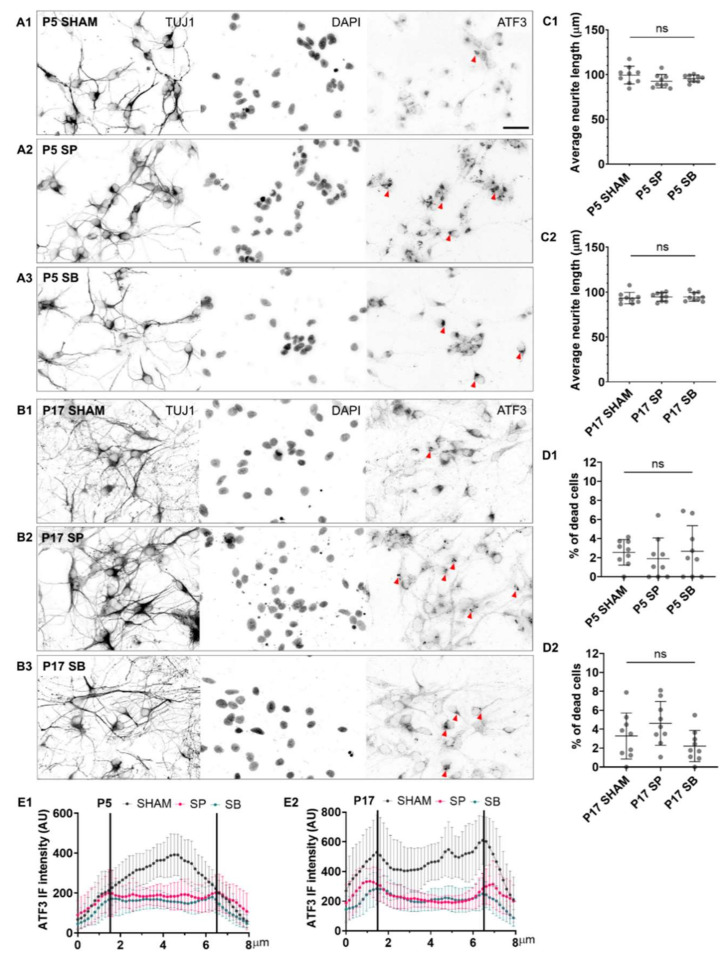
The effect of ATF3 inhibitors on intact neuronal networks. The primary neuronal cultures derived from the P5 (**A1**–**A3**) and P17 (**B1**–**B3**) cortex of *M. domestica* were treated at DIV4 with the MAPK inhibitors, JNK/c-Jun inhibitor SP600125 (SP), or the p38 inhibitor SB203580 (SB) for 24 h. The cells were fixed and stained at DIV5. Neurons were stained for β-tubulin III (TUJ1, left panel) and ATF3 (right panel). Red arrowheads indicate ATF3 cytoplasmic aggregates. Nuclei were counterstained with DAPI (middle panel). For a clearer visualization, the images are shown in the grayscale. Scale bar is 50 μm. (**C1**) The scatter plot shows the average neurite length of P5 cortical neurons for the experimental conditions shown from (**A1**–**A3**). Data are shown as mean ± SD. The total number of neurites analyzed P5 SHAM 184; SP 154; SB 216. One-way ANOVA followed by Holm-Šídák multiple comparisons test. P5 SHAM vs. P5 SP *p* = 0.13 ns; P5 SHAM vs. P5 SB *p* = 0.29 ns. (**C2**) The scatter plot shows the average neurite length of P17 cortical neurons for the experimental conditions shown from (**B1**–**B3**). Data are shown as mean ± SD. The total number of neurites analyzed P17 SHAM 215; SP 223; SB 209. One-way ANOVA followed by Holm-Šídák multiple comparisons test. P17 SHAM vs. P17 SP *p* = 0.82 ns; P17 SHAM vs. P17 SB *p* = 0.82 ns. (**D1**) The scatter plot shows the percentage of dead cells with mean ± SD. The total number of cells analyzed: P5 SHAM 675; SP 533; SB 543. One-way ANOVA followed by Holm-Šídák multiple comparisons test. P5 SHAM vs. P5 SP *p* = 0.77 ns; P5 SHAM vs. P5 SB *p* = 0.91 ns. (**D2**) The scatter plot shows the percentage of dead cells with mean ± SD. Number of cells analyzed: P17 SHAM 960; SP 738; SB 840. One-way ANOVA followed by Holm-Šídák multiple comparisons test. P17 SHAM vs. P17 SP *p* = 0.36 ns; P17 SHAM vs. P17 SB *p* = 0.36 ns. For each condition, images of at least three random fields per sample from 3 culture preparations were analyzed. (**E1**) Line scan analysis of P5 primary neurons for conditions shown from (**A1**–**A3**) and (**E2**) P17 primary neurons for conditions shown from (**B1**–**B3**). Data are shown as an average ± SD. The graph represents the average ATF3 immunofluorescence intensity measured in 10–16 randomly selected neurons for each condition (nucleus is within the vertical lines on the plot).

**Figure 5 ijms-23-04964-f005:**
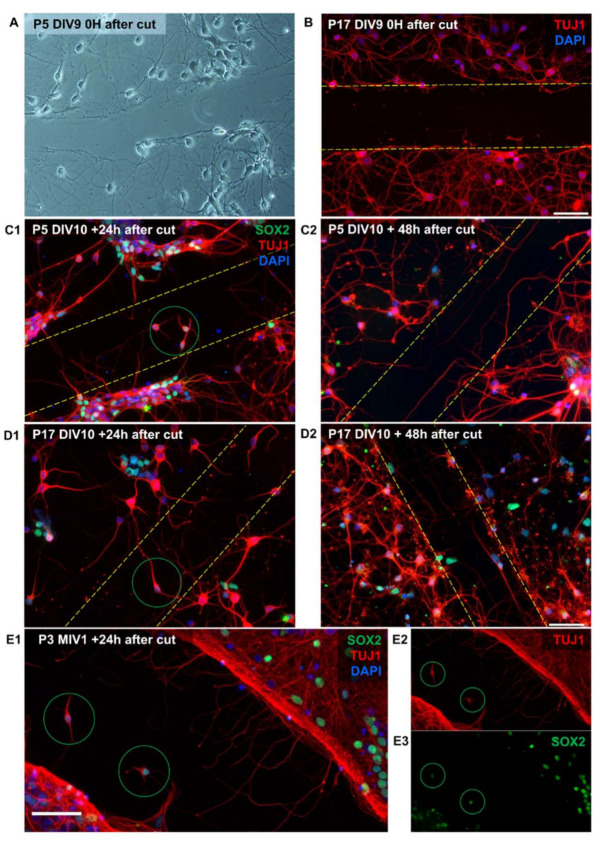
Cortical neuronal regeneration after injury in vitro. (**A**,**B**) Representative images of primary neuronal cultures immediately after a scratch. DIV9 live neurons from the P5 opossum cortex were imaged with phase contrast (**A**) and DIV9 neurons from the P17 opossum cortex fixed and stained for TUJ1 (red) and DAPI nuclear stain (blue) (**B**). (**C1**,**C2**) P5 and (**D1**,**D2**) P17 primary cortical neuronal cultures of *M. domestica* were cultured in neuronal medium, and the scratch was made at DIV10. Cells were fixed 24 h after cut (**C1**,**D1**) and 48 h after scratch (**C2**,**D2**) and immunostained for TUJ1 (red), SOX2 (green), and nuclei were counterstained with DAPI (blue). Notice new neurites crossing the scratched area. The scratch area is defined by the yellow dashed lines. Green circles indicate TUJ1+/SOX2+ immature neurons that have migrated to the scratch area. (**E1**–**E3**) P3 opossum cortical primary neuronal culture cultured for 1 month in vitro (MIV1). Scratch was made on MIV1, and cells were fixed 24 h later and immunostained for TUJ1 (red), SOX2 (green), and DAPI nuclear stain (blue). Note a substantial number of regenerating neurites crossing the scratch area. Green circles indicate TUJ1+/SOX2+ immature neurons that have migrated to the scratch area. Scale bar is 50 μm.

**Figure 6 ijms-23-04964-f006:**
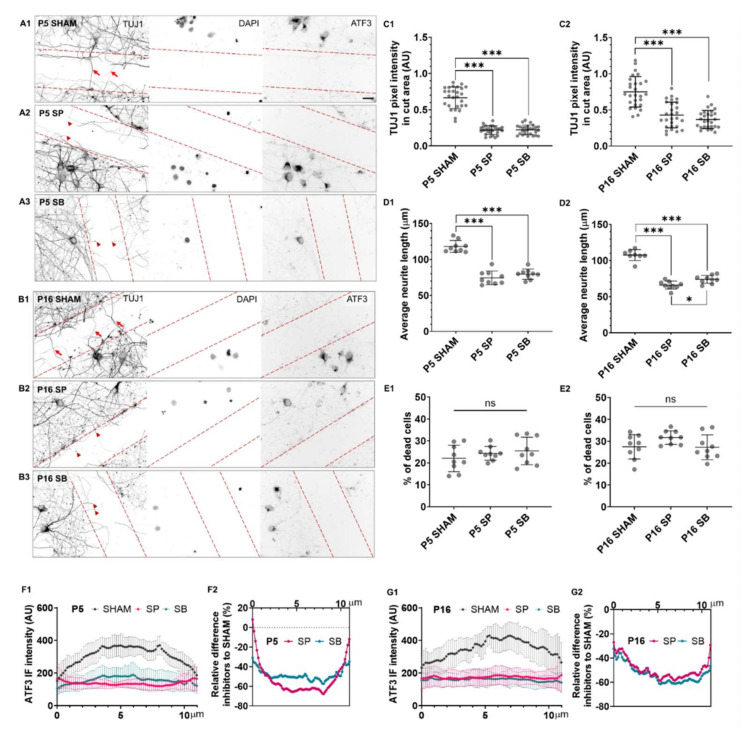
Reduction in neurite outgrowth by ATF3 inhibitors. Regrowth of TUJ1-positive neurites after scratch, with or without the presence of MAPK inhibitors SP600125 (SP, 50 μM) and SB203580 (SB, 1 μM). (**A1**–**A3**) P5 and (**B1**–**B3**) P16 primary cortical neuronal cultures of *M. domestica* were cultured for 2 weeks, and the scratch was made at DIV14. Inhibitors were added immediately over the next 24 h, and cells were fixed at DIV15 and immunostained for TUJ1 (left panels) and ATF3 (right panels). Nuclei were counterstained with DAPI (middle panels). The scratch area is defined by red dashed lines. On the left panels, red arrows indicate neurites crossing the area of the scratch, and red arrowheads indicate the bulb formation after the scratch. For a clearer visualization, the images are shown in grayscale. Scale bar is 20 μm, 40× magnification. (**C1**) The scatter plot shows P5 average TUJ1 pixel intensity for the experimental conditions shown from (**A1**–**A3**). Data are shown as mean ± SD. For each condition, 3 ROIs of 3 random fields per sample from 3 culture preparations were analyzed. Brown-Forsythe ANOVA followed by Dunnett’s T3 multiple comparisons test. P5 SHAM vs. P5 SP *p* < 0.001 ***; P5 SHAM vs. P5 SB *p* < 0.001 ***. (**C2**) The scatter plot shows P16 average TUJ1 pixel intensity for the experimental conditions shown from B1-B3. Data are shown as mean ± SD. For each condition, 3 ROIs of 3 random fields per sample from 3 culture preparations were analyzed. Brown-Forsythe ANOVA followed by Dunnett’s T3 multiple comparisons test. P16 SHAM vs. P16 SP *p* <0.001 ***; P16 SHAM vs. P16 SB *p* < 0.001 ***. (**D1**) The scatter plot shows the neurite lengths for P5 cortical neurons with mean ± SD. The total number of neurites analyzed P5 SHAM 335; SP 321; SB 312. One-way ANOVA followed by Holm-Šídák multiple comparisons test. P5 SHAM vs. P5 SP *p* < 0.001 ***; P5 SHAM vs. P5 SB *p* < 0.001 ***. (**D2**) The scatter plot shows the neurite length for P16 cortical neurons with mean ± SD. The total number of neurites analyzed P16 SHAM 213; SP 180; SB 236. One-way ANOVA followed by Holm-Šídák multiple comparisons test. P16 SHAM vs. P16 SP *p* < 0.001 ***; P16 SHAM vs. P16 SB *p* < 0.001 ***; P16 SP vs. P16 SB *p* = 0.01 *. (**E1**) The scatter plot shows the percentage of dead cells for P5 with mean ± SD. The total number of cells analyzed: P5 SHAM 704; SP 716; SB 663. One-way ANOVA followed by Holm-Šídák multiple comparisons test. P5 SHAM vs. P5 SP *p* = 0.38 ns; P5 SHAM vs. P5 SB *p* = 0.35 ns. (**E2**) The scatter plot shows the percentage of dead cells for P17 with mean ± SD. The total number of cells analyzed: P16 SHAM 457; SP 483; SB 479. One-way ANOVA followed by Holm-Šídák multiple comparisons test. P16 SHAM vs. P16 SP *p* = 0.15 ns; P16 SHAM vs. P16 SB *p* = 0.93 ns. For each condition, images of at least three random fields per sample from 3 culture preparations were analyzed. (**F1**) Line scan analysis of P5 primary neurons for the experimental conditions shown from (**A1**–**A3**) and (**G1**) P16 primary neurons for the experimental conditions shown from B1-B3 performed on an average intensity projection of a Z-stack. The line plot represents average ATF3 immunofluorescence intensity measured in 10–16 randomly selected neurons from each condition (nucleus is within the vertical lines on the plot). (**F2**,**G2**) The line plots represent a relative percentage difference in the average intensity of ATF3 immunofluorescence in neurons treated with inhibitor compared to the average intensity of ATF3 in neurons in the SHAM condition. Data are shown as an average ± SD.

**Figure 7 ijms-23-04964-f007:**
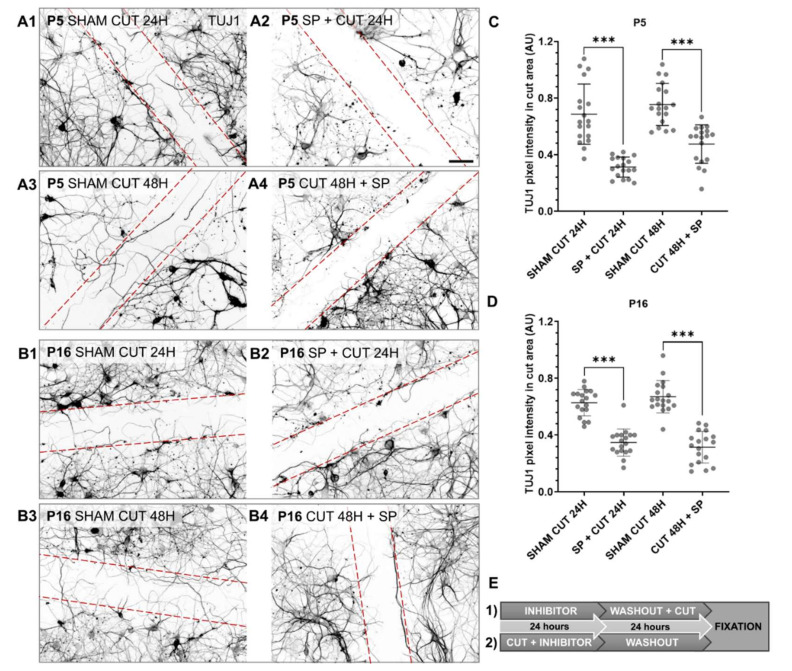
Prolonged effect of ATF3 inhibition on neurite outgrowth. (**A1**–**A4**) Cortical neurons derived from P5 opossum. (**A1**,**A3**) represent control neuronal cultures treated with PBS only. At DIV10, neurons were first treated with SP inhibitor for 24 h, followed by scratch and washout for the next 24 h (**A2**), or incubated with SP immediately after scratch over the next 24 h and followed by 24 h washout (**A4**). (**B1**–**B4**) Cortical neurons derived from P16 opossum. (**B1**,**B3**) represent control neuronal cultures treated with PBS only. At DIV10, neurons were first treated with SP inhibitor for 24 h, followed by scratch and washout for the next 24 h (**B2**), or incubated with SP immediately after scratch over the next 24 h and followed by 24 h washout (**B4**). Neurons were fixed and immunostained for TUJ1. The scratch area is defined by the red dashed lines. Scale bar is 50 μm. (**C**) The scatter plot shows P5 average TUJ1 pixel intensity. Data are shown as mean ± SD. For each condition, 2 ROIs of 3 random fields per sample from 3 culture preparations were analyzed. Welch’s *t*-test, SHAM CUT 24H vs. SP + CUT 24H *p* < 0.001 ***. Unpaired *t*-test, SHAM CUT 48H vs. CUT 48H + SP *p* < 0.001 ***. (**D**) The scatter plot shows P16 average TUJ1 pixel intensity. Data are shown as mean ± SD. For each condition, 2 ROIs of 3 random fields per sample from 3 culture preparations were analyzed. Unpaired *t*-test, SHAM CUT 24H vs. SP + CUT 24H *p* < 0.001 ***. Unpaired *t*-test, SHAM CUT 48H vs. CUT 48H + SP *p* < 0.001 ***. (**E**) Schematic representation of the time course of SP inhibitor application. (**1**) represents the application of SP for experimental conditions (**A2**,**B2**). (**2**) represents the application of SP inhibitor for experimental conditions (**A4**,**B4**).

**Figure 8 ijms-23-04964-f008:**
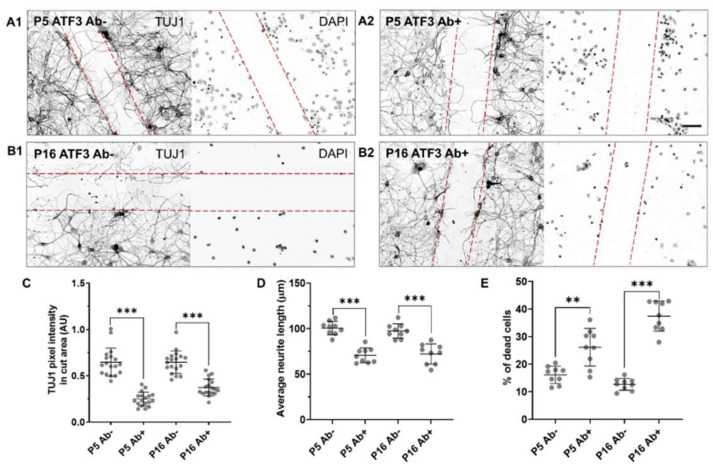
Inhibition of neuroregeneration by ATF3 antibody. (**A1**,**B1**) represents a control neuronal culture treated with PBS only. ATF3 antibody was added to P5 (**A2**) and P16 (**B2**) opossum cortical neurons in cell culture medium at DIV9 immediately after scratch. Neurons were fixed 24 h later and immunostained for TUJ1 (left panels). Nuclei were counterstained for DAPI (right panels). The scratch area is defined by the red dashed lines. Scale bar is 50 μm. (**C**) Scatter plot represents P5 and P16 average TUJ1 pixel intensities. Data are shown as mean ± SD. For each condition, 2 ROIs of 3 random fields per sample from 3 culture preparations were analyzed. Welch’s *t*-test, P5 Ab- vs. P5 Ab+ *p* < 0.001 ***. Unpaired *t*-test, P16 Ab- vs. P16 Ab+ *p* < 0.001 ***. (**D**) Scatter plot represents P5 and P16 average neurite length of TUJ1-positive neurons. Data are shown as mean ± SD. Number of neurites analyzed: P5 Ab- 270; P5 Ab+ 230; P16 Ab- 245; P16 Ab+ 214. Unpaired *t*-test P5 Ab- vs. P5 Ab+ *p* < 0.001 ***. Unpaired *t*-test, P16 Ab- vs. P16 Ab+ *p* < 0.001 ***. (**E**) Scatter plot represents the percentage of dead cells for P5 and P16 cortical cultures. Data are shown as mean ± SD. Number of cells analyzed: P5 Ab- 608; P5 Ab+ 784; P16 Ab- 516; P16 Ab+ 433. Welch’s *t* test P5 Ab- vs. P5 Ab+ *p* = 0.002 **. Welch’s *t* test, P16 Ab- vs. P16 Ab+ *p* < 0.001 ***. Images of 3 random fields were analyzed per sample from 3 culture preparations. The accepted level of significance was *p* < 0.05. *p* < 0.001 very significant ***, 0.001–0.01 very significant **, 0.01–0.05 significant *, ≥0.05 not significant (ns).

## Data Availability

The data sets used and/or analyzed during the current study are available from the corresponding author on reasonable request.

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
