# Peer review of "The Role of ATF3 in Neuronal Differentiation and Development of Neuronal Networks in Opossum Postnatal Cortical Cultures"

_ijms, 2022, doi:10.3390/ijms23094964_

Round 1

Reviewer 1 Report

Ijms-1681740. In this study, Petrovic et al. investigated the role of the transcription factor ATF3 in the development of cortical neurons of the opossum Monodelphis domestica. To do so, they carried out a comparative study on undifferentiated neurospheres vs. differentiating primary neurons taken from either P5-6 and P16-17 opossums, because the younger age group, but not the older one, display substantial neuroregenerative capacity after spinal cord injury. They provide a large body of molecular, morphological and functional (pharmacological) data which, they conclude, support the role of ATF3 in neurite outgrowth and regeneration capacity of younger neurons.

The study is well conducted and the abundant data provide strong support to the conclusions. It is overall well written. There are just some minor points which would benefit a revision, as follows:

Materials & Methods.

1) Line 139. Please, provide indication of the objectives used and software for image acquisition and analysis of scratch assay.

2) Line 141. Please, indicate the catalogue number (if any) for the SP and SB inhibitors purchased from SCBT.

3) Immunofluorescent staining. Lines 202, 203, 212. Please, provide the T°C used for antibody incubation.

4) Line 250. Please, name the company from which Trypsin was purchased and report the catalog n.

5) Statistics. Why not using two-way Anova for experiments such as those in Fig. 1 or Fig. 7B ?

Figures.

6) Figures are mispositioned throughout the text. Please, make sure that they are repositioned in the final version of the article.

7) In general, it is confusing to have figures lettering with letter+digit (e.g. A1, A2 etc..); it is generally preferred to have letters only (A, B, C, etc..).

8) Figure 1. Why there are no SD error bars? Why not using here scatter plots as in the other figures?

Results.

9) Lines 592-593. Please, indicate +/-SD corresponding to the data on TUJ1 pixel intensity.

10) It would be good if the authors could also provide data on the effects of overexpression of ATF3 (possibly modified with a Nuclear Localization Signal) to strengthen the conclusion (Line 715).“We assume that the localization of ATF3 in the nucleus keeps cells in a multipotent progenitor state and that this potential is attenuated after differentiation.”

Discussion.

10) Line 658 (development of neuronal polarity) The introduction reports that P0 opossums correspond to E12 rats, and P14 opossums to P0 rats, and this is fine. However, it would be useful for readers familiar with in vitro development of cortical and hippocampal neurons from mice and rats to have an idea if the primary neuronal cultures from opossum actually follow the same maturation trajectories of neuronal polarity and passes through the same developmental stages in vitro, as the rodents neuronal cultures (see for instance: http://dx.doi.org/10.1016/j.ceb.2012.05.011; http://dx.doi.org/10.3389/fncel.2014.00018; http://dx.doi.org/10.3389/fnins.2015.00116;  http://dx.doi.org/10.1016/j.mcn.2017.03.008 ). Henceforth, the authors are encouraged to consider adding to the Discussion a short paragraph to comment on this point.   

11) Line 677. The reference (48) doesn’t seem to be the correct one on this point.

12) Line 680. (migration of immature neurons to the injuried area). How can the authors exclude that glial cell are also participating to the invasion of  the scratched  region?

13) Line 689. (we established and described a new in vitro axotomy model). This seems a bit overstated. The model does not seem to be completely new, and how can the authors exclude that dendrites could also be trimmed ? Depending on the developmental stage, TUJ1 labels dendrites as well.

Very minor, general comment: there are different text types used throughout the manuscript.  Please, make sure that they are made consistent in the final version of the article.

Reviewer 2 Report

This is an interesting study that investigated the modulating effects of ATF3 on neural outgrowth, network formation and regeneration after injury. The authors demonstrated the involvement on ATF3 in this process with the pharmacologic inhibition of ATF3 reducing the capacity for neural outgrowth and regeneration. The study has a potential for high impact and is relevant to the journal. I have minor comments that the authors may wish to consider.

Minor comments:

  1. Any reason why the front size in figures 1 & 5 legends are different
  2. The ordering of the figures in the results are misplaced figure 3 comes first before figures 1 and 2, also figure 5 before 4 and 7 before 6.
  3. Can you provide the exact percentage instead of almost 25% or almost 20% increase in cell death?
